# Anti-Inflammatory and Immune Properties of Polyunsaturated Fatty Acids (PUFAs) and Their Impact on Colorectal Cancer (CRC) Prevention and Treatment

**DOI:** 10.3390/cancers15174294

**Published:** 2023-08-28

**Authors:** Alireza Tojjari, Khalil Choucair, Arezoo Sadeghipour, Azhar Saeed, Anwaar Saeed

**Affiliations:** 1Division of Hematology and Oncology, Department of Medicine, University of Pittsburgh Medical Center, Pittsburgh, PA 15213, USA; alirezatojjari@gmail.com; 2Division of Hematology and Oncology, Department of Medicine, Barbara Ann Karmanos Cancer Institute, Detroit, MI 48201, USA; choucairk@karmanos.org; 3Department of Biochemistry, Faculty of Biological Sciences, Tarbiat Modarres University, Tehran 14115-175, Iran; arezoo.sadeghipour@modares.ac.ir; 4Department of Pathology, Huntsman Cancer Institute, University of Utah, Salt Lake City, UT 84112, USA; Azhar.Saeed@uvmhealth.org; 5UPMC Hillman Cancer Center, Pittsburgh, PA 15232, USA

**Keywords:** colorectal cancer, polyunsaturated fatty acids, omega-3 fatty acid, cancer prevention, diet, cancer treatment

## Abstract

**Simple Summary:**

In our review, we investigate polyunsaturated fatty acids, or PUFAs for short, and their potential role in preventing and treating colorectal cancer (CRC). We are inspired to do this research because CRC is becoming more common, and we are always looking for new ways to tackle it. We aim to shed light on how PUFAs, known for their anti-inflammatory and immune-boosting properties, might influence CRC. By understanding more about how PUFAs interact with CRC on a molecular level, we could open up new dietary strategies and treatments. This could mean better patient outcomes and a powerful new tool in our fight against CRC.

**Abstract:**

Colorectal cancer (CRC) remains a leading cause of death from cancer worldwide, with increasing incidence in the Western world. Diet has become the focus of research as a significant risk factor for CRC occurrence, and the role of dietary polyunsaturated fatty acids (PUFAs) has become an area of interest given their potential role in modulating inflammation, particularly in the pro-carcinogenic inflammatory environment of the colon. This work reviews the main types of PUFAs, their characteristics, structure, and physiologic role. We then highlight their potential role in preventing CRC, their signaling function vis-à-vis tumorigenic signaling, and their subsequent potential role in modulating response to different treatment modalities. We review pre-clinical and clinical data and discuss their potential use as adjunct therapies to currently existing treatment modalities. Given our understanding of PUFAs’ immune and inflammation modulatory effects, we explore the possible combination of PUFAs with immune checkpoint inhibitors and other targeted therapies.

## 1. Background

Colorectal cancer (CRC) remains a leading cause of cancer death globally, accounting for around 10% of all cancer-related deaths in developed nations [1]. While increased incidence can be attributed to longer life expectancy, lifestyle, and diet, intake has become the research focus as a significant risk factor for CRC occurrence [2].

Like other chronic degenerative illnesses, tumorigenesis appears to be partially mediated via pro-inflammatory mechanisms that alter the human immune system, adipokine production, and metabolic pathways. Recently, dietary polyunsaturated fatty acids (PUFAs) have become an area of interest, given their potential role in modulating inflammation, particularly in the pro-carcinogenic inflammatory environment of the colon [3]. PUFAs are classified based on the number of carbon atoms after the final double bond in the fatty acid chain; Omega-3 (ω-3) fatty acids such as docosahexaenoic acid (DHA) and eicosapentaenoic acid (EPA) contain n-3 subsequent carbons, while omega-6 (ω-6) fatty acids such as arachidonic acid (AA) and linoleic acid (LA) contain n-6 carbon atoms [4,5].

The Western diet, rich in processed meat and high-fat content food, has a high composition of ω-6 and has been established as a risk factor for CRC [6]. An opposite trend was observed with diets rich in olive oil and other fish-based diets, even when consumed as part of a 20% high-fat diet [7,8,9,10,11]. Interesting results also included decreased tumor incidence in animals with high fish-oil diets compared to low-fat (5%) diets, including animal fat and vegetable oil. Unlike most vegetable oils, mainly composed of the ω-6 PUFA LA, fish oil has a relatively high concentration of the ω-3 PUFAs EPA and DHA [11]. As a reservoir for bioactive chemicals, n-3 PUFAs also affect membrane architecture, gene and protein expression, activity, and stability [12]. Additionally, n-3 PUFAs resolve inflammation by producing tissue-regenerating protectins and resolvins. It is thought that n-3 PUFAs, which are anti-inflammatory, and n-6 PUFAs, which are pro-inflammatory, have opposing effects on the inflammatory process [13].

While many epidemiological and experimental animal model studies have found a positive link between dietary fat consumption and the development of various types of cancer, n-3 PUFAs have generated considerable interest as nutritional supplements due to their potential anti-inflammatory properties, anticancer effects, and cancer-preventive effects [14,15,16,17]. Despite supportive pre-clinical data, no strong evidence-based and definitive dietary advice or clear clinical guidelines for n-PUFA consumption exists. This review provides a comprehensive overview of the landscape of PUFA, specifically ω-3, and its potential role in CRC prophylaxis and therapy.

## 2. Polyunsaturated Fatty Acids (PUFAs)

Fatty acids (FAs) are hydrocarbon chains with carboxyl and methyl groups at opposite ends. Three primary types of FA exist and differ based on their saturation levels: saturated fatty acids (SFAs), monounsaturated fatty acids (MUFAs), and polyunsaturated fatty acids (PUFAs) [18]. SFAs have no double bonds in their carbon atoms, while MUFAs have only one, and PUFAs have two or more [19]. The three types further harbor various physiological effects. For example, the consumption of SFA has been associated with the onset of metabolic dysfunction. The consumption of MUFAs and PUFAs, on the other hand, has beneficial effects on metabolism [20]. Many PUFAs exist, but ω-3 and ω-6 PUFAs are considered essential, as the human body lacks ω-3 desaturase, an enzyme that is necessary for the synthesis of α-linoleic acid (ALA; 18:3n-3) and linoleic acid (LA; 18:2n-6). Thus, humans cannot efficiently produce either PUFAs independently. Although humans cannot produce these PUFAs independently, they may undergo further elongation and desaturation throughout the metabolic process [21].

The main ω-3 PUFAs, EPA and DHA, are primarily found in oily, cold-water fish like mackerel and are often considered to be the most bioactive of the ω-3 species; however, docosapentaenoic acid (DPA), an intermediate of EPA and DHA, may also have beneficial health effects [20,22]. The ω-6 PUFAs are essential fatty acids with different metabolic characteristics compared to ω-3 PUFAs.

Different structural features of these FAs lead to distinct functional differences in how they influence inflammation and metabolism [23]. For instance, a higher risk of cardiovascular disease is linked to consuming saturated FAs because of their pro-inflammatory effects. Conversely, when consumed regularly, ω-3 PUFAs may help reduce inflammation and have been linked to a reduced risk of cardiovascular disease.

## 3. CRC Prevention

Effective CRC prevention methods include (i) health education promoting behavior modifications such as weight management and increased physical activity; (ii) screening initiatives for high-risk populations; and (iii) endoscopic monitoring. CRC prevention approaches aim to identify and remove asymptomatic colorectal adenomas in healthy individuals either directly through colonoscopy or flexible sigmoidoscopy (FS) screenings or indirectly via colonoscopies prompted by fecal occult blood tests. These tactics are rooted in the lengthy natural history of “sporadic” colorectal carcinogenesis in humans, encompassing the formation and growth of benign adenomas or polyps before their progression into clinically noticeable malignant adenocarcinoma [24,25]. The mortality rate associated with CRC can be reduced by performing colonoscopies and excising polyps, as evidenced by a 50% reduction in risk found in the US National Polyp Study [26].

Various potential CRC chemo-preventive agents, including non-steroidal anti-inflammatory drugs (NSAIDs), hormone replacement therapy, and the administration of micronutrients such as folic acid and vitamin D, have been studied [27]. Among NSAIDs, aspirin has the most vital support for its role in colorectal cancer chemoprevention [28]. However, due to the small risk of gastrointestinal and intra-cerebral bleeding, as well as the ongoing debate surrounding the ideal daily dosage and the absence of a well-defined at-risk population for whom the benefits would outweigh the risks, aspirin has not been recommended for primary or secondary CRC chemoprevention [28]. Recently, researchers have started investigating combinational approaches for effective CRC prophylaxis, including Omega-3 fatty acids, which will be discussed in the next section.

## 4. Effects of ω-3 PUFAs on CRC Prophylaxis

Numerous clinical studies have established a connection between specific fatty acids and the risk of colorectal cancer [29]. Research involving animal models and cell cultures has revealed that ω-3 PUFAs exhibit anti-CRC properties through several proposed mechanisms. These include (i) the regulation of cyclooxygenase (COX) activity, which is involved in inflammation; (ii) changes in cell membrane dynamics and receptor function, which can impact cellular signaling; (iii) an increase in cellular oxidative stress, which may lead to the destruction of cancer cells; and (iv) the production of new anti-inflammatory lipid compounds derived from EPA and DHA. These compounds, such as resolvins, protectins, and maresins, contribute to the resolution of inflammation and promote tissue repair [30] (Figure 1).

Recent studies suggest that a high intake of marine-based ω-3 PUFAs is associated with a reduced risk of colorectal adenoma [31]. However, some human observational studies have yielded inconsistent results regarding the benefits of ω-3 PUFA consumption. This inconsistency may be attributed to methodological challenges in measuring ω-3 PUFAs or fish intake and the possibility that moderate dietary ω-3 PUFA exposure might not consistently provide anti-CRC effects. Consuming two to three servings of oily fish per week equates to a daily dose of about 500 mg of EPA and DHA [22]. To ensure the safe and effective administration of high doses of EPA, a 500 mg capsule containing EPA in its free fatty acid (FFA) form is currently available, allowing daily doses of up to 2 g. Optimal absorption of EPA occurs in the small intestine, where it is released from the capsules, thus minimizing gastrointestinal side effects. The FFA form of EPA has significantly higher bioavailability compared to the more common ethyl ester or triglyceride forms of EPA [32].

### 4.1. Diet and ω-3 Content

Diet has a significant role in developing and spreading chronic diseases, including inflammation and cancer [33]. It is currently accepted that inflammation is a cancer risk factor that may promote the growth of many tumors and that an inflammatory microenvironment is a crucial feature of every tumor [34,35]. Several observational studies have investigated the relationship between ω-3 fatty acid intake and CRC risk in combination with consuming macronutrients like fiber and micronutrients like vitamin C. A single research study involving 44,039 participants reported a connection between consuming ω-3 fatty acids and an elevated risk of digestive cancers (encompassing colon, rectum, stomach, liver, esophagus, and pancreas) among individuals who had a below-median intake of fruits and vegetables and below-median intake of vitamin C [36].

### 4.2. The Role of ω-3 and Fiber Intake in CRC Prophylaxis

Based on a meta-analysis of 21 studies conducted in 2019 [37], no direct relationship was discovered between fiber intake and the risk of CRC. However, there was significant variation among the studies. While most studies from the US did not report any association [38,39], the European Prospective Investigation into Cancer and Nutrition cohort consistently found that consuming fiber is linked to a lower risk of CRC [40,41].

Recently, two studies investigated the relationship between ω-3 FAs, dietary fiber consumption, and CRC risk. The first research study included 4967 patients, 222 of whom had CRC. This research demonstrated that consuming ω-3 FAs, especially from non-marine sources, was associated with an increased risk of CRC in those with a fiber consumption below the median. The CRC risk ratio was 1.96 (95% CI: 1.20–3.19), and there was a statistically significant positive interaction between dietary fiber and ω-3 FA consumption and CRC risk. The more substantial sample size of 134,017 participants in the second research study included 1952 individuals with CRC. The study found that greater ω-3 FA consumption was related to a decreased risk of colorectal cancer [42,43].

The potential anticancer effects of fiber may be linked to its impact on the gut microbiota. When bacteria ferment fiber, they generate short-chain fatty acids (SCFAs), which have been shown to modulate the immune system and metabolism and lower the risk of colorectal cancer [44]. SCFAs exhibit direct anti-CRC activity within the colorectum by triggering the apoptosis of epithelial cells. Additionally, SCFAs regulate the host’s anti-tumor immune response, which could explain why ω-3 FAs are more effective in protecting against tumors with deficient DNA mismatch repair (dMMR) [45]. Several cross-sectional studies have observed a positive correlation between increased fiber consumption, higher fecal levels of SCFAs, and a greater abundance of SCFA-producing bacteria, including Eubacterium rectale, Roseburia species, and F. prausnitzii. However, not all studies have reported the same association [46].

### 4.3. The Role of Fish Oil (FO)

Fish oil (FO) supplements have gained considerable popularity, with the prevalence of regular usage reported between 31.2% and 32.6% among 427,678 participants who reported habitual use of fish oil supplements in a cohort study [47]. Although some studies have incorporated FO supplement data within the overall evaluation of ω-3 FA consumption, only a handful have specifically investigated the impact of FO supplements on CRC risk [36,48,49,50]. This distinction is crucial, as FO supplements may have distinct effects on CRC risk compared to dietary ω-3 FA intake from other sources. One study that analyzed FO supplements independently reported that individuals who consumed FO supplements for more than three days per week over a period of at least three years exhibited a 49% reduction in CRC risk compared to non-users [51]. However, this finding did not achieve statistical significance, potentially due to the study’s statistical power limitations.

### 4.4. Combining EPA and Aspirin

The exact mechanisms through which EPA and aspirin exert their anti-CRC effects remain elusive. One prevailing hypothesis suggests that their modulation of cyclooxygenase (COX) activity primarily contributes to their antineoplastic properties. However, it is also postulated that these compounds may act through COX-independent pathways to exert their anti-CRC effects. EPA and aspirin are both known to be effective COX-1 inhibitors, but they modulate COX-2 activity in distinct ways, forming different bioactive lipid mediators. For instance, EPA is responsible for the synthesis of prostaglandin E3 (PGE3), whereas aspirin promotes the production of 15-hydroxyeicosatetraenoic acid (15R-HETE) [52,53]. The unique lipid mediators generated through these pathways are thought to play a crucial role in these compounds’ anti-inflammatory and antineoplastic activities. Another potential mechanism through which aspirin may influence the anti-inflammatory properties of EPA is by irreversibly acetylating the COX enzymes that convert EPA to 18R-hydroxyeicosapentaenoic acid (18R-HEPE) [54]. This reaction forms a potent anti-inflammatory molecule, resolvin (Rv) E1 or trihydroxy-EPA. This interaction between EPA and aspirin provides a biological basis for their potential synergistic effects in combating CRC. Support for the synergistic effects of aspirin and ω-3 PUFAs in preventing CRC comes from several ex vivo human platelet aggregation studies. These experiments have consistently demonstrated that combining aspirin and ω-3 PUFAs exhibits additive and synergistic effects on platelet aggregation inhibition, which may contribute to their overall anti-CRC activity [55,56].

In summary, the precise mechanisms through which EPA and aspirin exert their anti-CRC effects still need to be fully understood. However, current evidence points to their modulation of COX activity and the formation of distinct bioactive lipid mediators as crucial contributing factors. Additionally, the potential interaction between EPA and aspirin in forming anti-inflammatory molecules such as RvE1 may underlie their synergistic effects in preventing CRC. Further research is warranted to elucidate the complex interplay between these compounds and their anti-CRC mechanisms of action.

## 5. The Impact of ω-3 on CRC-Related Molecular Signaling Pathways

ω-3 FAs have been found to downregulate the activity of signaling networks that promote CRC, including the Wnt/ß-catenin, MAPK/ERK, PI3K/AKT/Bcl-2, and PI3K/PTEN pathways. These FAs have been shown to integrate into the plasma membrane of cancer cells, alter their composition and fluidity, and by doing so, they prevent signal transduction and reduce cancer cells’ ability to survive and increase apoptosis [57] (Figure 2).

The Wnt/ß-catenin signaling pathway is one of the pathways that control apoptosis in CRC cells [58]. It is well demonstrated in vitro (HCT116, SW480, and Caco-2 CRC cell lines) that DHA can trigger apoptosis and avoid improper regulation of β-catenin protein by lowering the nuclear and total amounts of this protein in the human colon cancer cells and inhibiting COX-2 [58,59]. Treatment with DHA also reduced the levels of the anti-apoptotic protein survivin and other TCF-β-catenin target gene products, such as peroxisome proliferator-activated receptor-d, MT1-MMP, MMP-7, and VEGF, implicated in controlling apoptosis, tumor invasion, and neo-angiogenesis [58].

AKT, activated through PI3 kinase, is crucial in cellular processes such as proliferation, differentiation, metabolism, and apoptosis. DHA triggers apoptosis in CaCo-2 cells by enhancing caspase-3 activation and poly-ADP-ribose polymerase cleavage. Short-term DHA supplementation elevates AKT phosphorylation at Ser473 and Thr308 residues, while long-term supplementation inhibits Ser473 phosphorylation, potentially reducing its activity. Given AKT’s involvement in cell survival, restricting its total activity could lead to apoptosis. DHA might also influence the upstream signaling pathways of MEK, ERK1/2, or p38 MAPK in activating PDK1, the kinase responsible for Thr308 phosphorylation. The increase in ERK1/2 phosphorylation upon DHA supplementation may indicate PKC activation, as PKC can activate the ERK cascade through Raf-1. A delayed stress response due to DHA supplementation could help suppress AKT signaling, thereby inducing apoptosis via ASK1 upstream of p38 MAPK. It is possible that DHA supplementation does not directly activate ASK1 but rather affects other pro-apoptotic proteins such as Fas and TRAF2. Oxidative stress caused by PUFA supplementation may result in DNA damage and P53 phosphorylation at Ser15, with p38 MAPK playing a role in this process. Overall, DHA induces apoptosis in cancer cells by inhibiting survival-related kinases, suppressing AKT Ser473 phosphorylation, and promoting p38 MAPK phosphorylation [60].

PTEN, a lipid phosphatase, is responsible for the dephosphorylation of PIP3 and acts as a critical inhibitory component of the PI3K/AKT signaling pathway. Phosphorylation of PTEN is inhibitory and results in its inactivation as a negative regulator of the PI3-K pathway. DHA effectively lowers PTEN phosphorylation, augmenting its inhibitory effect on the PI3-K pathway. A strong correlation exists between PTEN absence and AKT activation in tumor cell lines. As described earlier, AKT is a crucial target of PI3-K and governs a wide range of proteins involved in cell survival and death decisions through phosphorylation. A colon cancer cell model study demonstrated that DHA treatment significantly reduced AKT phosphorylation. DHA treatment leads to AKT inactivation, decreasing BAD and FKHR phosphorylation. Consequently, this promotes apoptosis, as evidenced by elevated caspase-3 and PARP cleavage [61].

Huang et al. showed that ω-3 FAs reduced MNU-induced CRC in rats and inhibited AKT/Bcl-2 signaling, leading to the prevention of CRC cell colony formation and invasion, decreasing cell proliferation and increasing apoptosis [62].

## 6. Role of ω-3 FAs in Patients with An Established CRC Diagnosis

Several randomized controlled trials (RCTs) and clinical studies have investigated the effectiveness of ω-3 FAs supplements in patients with CRC. These studies have covered various aspects of CRC management, including (i) pre-and postoperative care, (ii) cancer cachexia treatment, and (iii) chemotherapy-induced peripheral neuropathy management. In these trials, ω-3 FAs were delivered in a variety of forms, such as capsules, oral nutritional supplements (ONS), nasogastric enteral feeds (EN), and parenteral nutrition (PN) [63]. While capsules or tablets are often easier to tolerate than ONS, ONS provide essential calories and protein, crucial for patients suffering from cancer-related cachexia. In general, RCTs have shown that interventions using ω-3 FAs capsules are associated with higher adherence rates than other interventions, such as ONS, EN, and PN [64,65].

Exploring the influence of marine ω-3 fatty acid supplements on the evolution of colorectal cancer provides insights into their potential role in modulating immune and inflammatory reactions. Song et al.’s research did not establish a strong link between ω-3 supplementation and the risk of colorectal polyps. However, deeper analysis unveiled some noteworthy patterns. Specifically, among African Americans, there was a suggestive trend where ω-3 supplementation seemed to lower the risk of adenomas, hinting at a connection between ω-3 fatty acids and immune system adjustments. This observation complements the growing body of knowledge about the role of ω-3 in regulating immune responses and inflammation in the context of cancer development [66].

While the study’s results in individuals with typical risk did not provide conclusive evidence, they highlight the nuanced relationship between ω-3 fatty acids, immune reactions, and colorectal well-being. To fully grasp the racial disparities and the interplay with baseline ω-3 concentrations, more in-depth studies are essential. This will help demystify the intricate dynamics between ω-3 fatty acids and the immune-inflammatory milieu in colorectal tumor development. In a broader context, for the average-risk American demographic, a daily dose of 1g marine ω-3 fatty acids might not curtail the risk of early-stage colorectal lesions. However, exceptions might exist for individuals with inherently low ω-3 levels or those of African American descent, as observed in a large-scale randomized clinical trial involving 25,871 participants [66].

Investigating the effects of fish oil supplementation, the study conducted by Anti et al. involved a treatment arm comprising 10 participants who were administered fish oil abundant in eicosapentaenoic acid (4.1 g/day) and docosahexaenoic acid (3.6 g/day). This intervention yielded a remarkable decrease in the average percentage of replicative “S”-phase cells situated in the upper segment of colonic crypts, a parameter acknowledged for its reliability in assessing the risk of colon cancer. Notably, this reduction attained statistical significance as early as 2 weeks after the commencement of treatment, persisting consistently over the entire 12-week study duration. In stark contrast, no observable modifications surfaced in the upper-crypt labeling of the placebo group, encompassing 10 participants [67].

A recent survey on the Nurses’ Health Study (NHS) and Health Professionals Follow-up Study (HPFS) cohorts assessed the relationship between dietary ω-3 FA intake and CRC outcomes. The study revealed that CRC patients who consumed higher amounts of ω-3 FAs had a lower risk of CRC-specific mortality [48]. One possible explanation is that ω-3 FAs may enhance the effectiveness and safety of chemotherapy and radiation therapy, alleviate cancer-related wasting syndrome, and exert a direct anti-CRC effect [64]. Analysis of two RCTs of adjuvant chemotherapy revealed a connection between high intake of ω-3 FAs and outcomes of CRC. For example, In a 7-year trial with 89,803 participants, those who ate dark fish had superior disease-free survival, recurrence-free survival, and overall survival than those who did not [68]. A summary of clinical and pre-clinical research on CRC using ω-3 FAs is presented in Table 1.

### 6.1. Perioperative ω-3 FAs and Postoperative Outcomes in CRC Patients

Studies have investigated the impact of enteral and parenteral supplementation of ω-3 FAs and other nutrients on clinical outcomes such as postoperative complications, infections, weight, length of stay, and proinflammatory cytokine levels in individuals having undergone primary CRC resection. According to a meta-analysis conducted in 2016, ω-3 FA-enriched nutrition significantly reduced infectious complications, hospital stay, and plasma levels of interleukin 6 (IL-6) and tumor necrosis factor-alpha (TNF-α) in CRC patients. The ω-3 FAs doses administered ranged from 0.05 g/kg body weight to 3.3 g/L in PN feed, and the duration of treatment varied from 5 days before surgery to 14 postoperative days [84]. On the other hand, a meta-analysis conducted in 2020 showed no notable distinction in body mass index (BMI), serum albumin level, wound infections, or pneumonia among postoperative GI cancer patients who were administered enteral nutritional therapy that included ω-3 FAs [80].

Three RCTs have investigated the effectiveness of ω-3 FA-containing ONS in patients undergoing primary CRC surgery, using varying doses and durations. One RCT in Denmark evaluated preoperative ONS and observed an increase in EPA levels and production of the anti-inflammatory EPA-derived leukotriene B5 (LTB5), with a decrease in AA and leukotriene B4 (LTB4) levels on the first postoperative day. However, there was no significant difference in clinical outcomes, such as postoperative complications, between the treatment and control groups. Long-term follow-up revealed no overall survival benefit and an increased risk of all-cause mortality in the treatment group after adjusting for age, disease severity, and use of adjuvant chemotherapy [82]. The other two RCTs did not show a significant difference in the length of hospital stay or postoperative complications. Still, the investigators did observe a significant increase in postoperative weight recovery in the treatment group [85,86].

RCTs have also evaluated PN-containing ω-3 FAs in patients undergoing primary CRC resection. The results of these trials have been mixed. One study found no significant reduction in postoperative complications but did report a decrease in systemic inflammatory response syndrome [87]. However, another study found an increased rate of anastomotic leaks and urinary tract infections compared to saline administration. Two other RCTs found no significant differences in immune response markers such as IL-6, CRP, TNF-α, or neutrophil phagocytosis index in the ω-3 FA-containing PN arm [88]. An ongoing RCT is assessing the primary outcome of changes in phagocytosis of pathogens, specifically Escherichia coli, using a mixed ω-3 FA-containing oral nutritional supplement for 5 days before laparoscopic CRC resection [89,90]

### 6.2. Cancer Cachexia

The use of ω-3 FA supplementation for cancer-related cachexia has been extensively studied. Participants in these studies were predominantly diagnosed with cancers affecting the pancreas, gastrointestinal tract, or lungs [91]. Some studies on pancreatic cancer suggest that ω-3 PUFA supplementation could benefit patients with upper gastrointestinal and pancreatic solid tumors by improving weight gain, appetite, and overall quality of life and reducing postoperative complications [92]. The survey on CRC patients demonstrated that, during the ω-3 PUFA supplementation period before chemotherapy, participants experienced a significant weight gain (average increase of 2.5 kg), even though their total protein and energy intake decreased. Importantly, this weight gain persisted even after the initiation of chemotherapy [93]. Another RCT was conducted to investigate the effects of EPA-enriched enteral nutrition in patients undergoing oesophagectomy for cancer. The results showed that patients who received EPA-enriched enteral nutrition for 5 days preoperatively and 21 days postoperatively preserved their fat-free mass compared to those who received standard enteral nutrition. The total weight loss in the EPA group was 1.2 kg (vs. 1.9 kg in the control). Moreover, the EPA group demonstrated a reduced stress response compared to the control group, as measured by serum TNF-α, IL- 8, and IL- 10 levels. However, the two groups had no significant difference in postoperative complications [94].

### 6.3. Ω3 FAs: An Adjunct to Chemotherapy?

Several studies have assessed ω-3 FA treatment as a complementary therapy to conventional systemic therapies, examining various outcomes such as treatment tolerability, quality-of-life measures, and the systemic inflammatory response.

An RCT was conducted on patients with CRC receiving chemotherapy to assess the efficacy of a specific probiotic (Hexbio) and ω-3 FAs (EPA and DHA) over 8 weeks. The double-blind and placebo-controlled study reported improved overall health and reduced chemotherapy-related adverse events such as diarrhea and anorexia. The study also showed decreased IL-6 levels but no significant difference in TNF-α or CRP levels [95]. Another RCT found that ω-3 FAs capsules significantly reduced the Glasgow Prognostic Score (GPS), a prognostic cancer marker based on inflammation composed of CRP and albumin, in patients with local advanced rectal cancer receiving neoadjuvant radiotherapy [96]. In a double-blind, placebo-controlled, randomized controlled trial, the effectiveness of ω-3 FAs in mitigating oxaliplatin-induced peripheral neurotoxicity (OXIPN) was investigated: ω-3 FA capsules (640 mg, three times daily) containing DHA (54%) and EPA (10%) reduced both the frequency and severity of OXIPN. A recent in vitro investigation revealed a unique potential mode of action, demonstrating that a blend of EPA, grape seed extract, and epigallocatechin-3-gallate (present in green tea) could effectively inhibit mTOR signaling, similar to rapamycin’s impact on SW480 and HCT116 cell lines [97].

## 7. Immune Modulation by ω-3 FAs in CRC

O3FAs have gained prominence in CRC research due to their potent anti-inflammatory properties. Their ability to bolster immune function is crucial for both the prevention and treatment of CRC.

Upon interaction with cell surface G protein-coupled receptors (GPCRs), ω-3 FAs act as anti-CRC agents by triggering pro-apoptotic signaling. Notably, these GPCRs are found on non-epithelial cells like adipocytes and macrophages. Their activation can alter macrophage polarization, potentially reducing inflammation—a key factor in ω-3 FAs’ anticancer effects [98].

A meta-analysis revealed that CRC patients who received ω-3 FA supplementation exhibited reduced levels of specific inflammatory mediators [99]. In vitro research identified DHA’s role in inhibiting CRC growth and promoting apoptosis through autocrine production of tumor necrosis factor-alpha (TNFα) via microRNA (miR)-21 [100]. Omega-3 FAs can modulate gene expression through epigenetic mechanisms, such as DNA methylation and miR gene regulation, influencing colorectal carcinogenesis processes [3,101].

Song et al. found a link between marine ω-3 FA intake and reduced CRC risk in cases with high FOXP3 + T-cell infiltration. They suggested that marine ω-3 FAs might enhance the immune response against cancer by regulating FOXP3 + Treg cells [102].

Another study showed that ω-3 supplementation, especially when combined with vitamin D, significantly lowered inflammatory markers and tumor markers [77]. DHA was found to mitigate inflammation in the early stages of the AOM/DSS colitis-associated colon cancer mouse model, inhibiting subsequent tumor formation [98].

A 2023 meta-analysis by Haoshuang Liu et al. emphasized the benefits of O3FA supplementation around the time of CRC surgery, leading to reduced inflammatory markers and shorter hospital stays [103]. Flaxseed oil, rich in omega-3 PUFAs, was highlighted for its potential health benefits, including its anticancer properties [104].

Moreover, research centered on methotrexate (MTX), a formidable anti-cancer medication, unveiled the protective qualities of long-chain polyunsaturated fatty acids (LC-PUFAs) against the renal toxicity induced by MTX. The findings posited that LC-PUFA might act as a valuable co-agent with MTX, tempering its detrimental kidney effects via its antiapoptotic, antioxidant, and anti-inflammatory mechanisms [105].

In summary, omega-3 fatty acids, particularly O3FAs, have shown promising results in CRC management. Their anti-inflammatory and immune-boosting properties make them potential adjuncts in CRC treatment, offering hope for improved patient outcomes.

## 8. Adverse Effects of ω-3 FAs

The duration of ω-3 FA supplementation in clinical studies varies considerably, lasting from as little as four weeks to six months. It has also been observed that the tissue incorporation of ω-3 FAs in humans plateaus after a few weeks [106]. However, the most effective duration of ω-3 FA treatment to achieve maximum antineoplastic effects is yet to be determined [106].

In general, ω-3 FA formulations are well tolerated. The most common adverse effects when given in high doses are gastrointestinal disturbances, with diarrhea being the most commonly reported symptom [107]. Other side effects include nausea, abdominal pain/discomfort, constipation, and vomiting. Non-gastrointestinal side effects include nasopharyngitis, arthralgia, headache, pruritus, rash, and dysgeusia [108].

## 9. Future Perspective

Despite abundant observations regarding the potential role of ω-3 FAs in CRC prevention, therapy, and supportive care management of CRC patients, the exact role and position of ω-3 FAs, such as EPA and DHA, in the landscape of CRC management remains to be elucidated.

For example, given ω-3 FAs’ immune and inflammation-modulatory properties, opportunities exist to explore the preventive role of long-term ω-3 FA supplementation, especially in patients with known family histories of CRC or pre-disposing syndromes such as Lynch. Similarly, combining ω-3 FAs with immune checkpoint inhibitors may be a route to be explored, given the immune-modulatory effect ω-3 FAs can play within the tumor microenvironment. Lastly, given the role ω-3 FAs play in regulating pro-tumorigenic signaling pathways, options to combine ω-3 FAs with targeted therapies could also be explored.

Other areas remain to be explored, including defining and implementing standardized ω-3 FA composition and dosing to ensure accurate and reproducible results across studies. Similarly, the optimal dosage and formulation patients can tolerate must be determined.

## 10. Conclusions

In conclusion, a deeper understanding of the relationship between ω-3 FAs and CRC may lead to the development of innovative preventive and therapeutic strategies to enhance current treatment options. By considering host characteristics, tumor types, and economic factors in future clinical trials, researchers can pave the way for more personalized and effective CRC care.

## Figures and Tables

**Figure 1 cancers-15-04294-f001:**
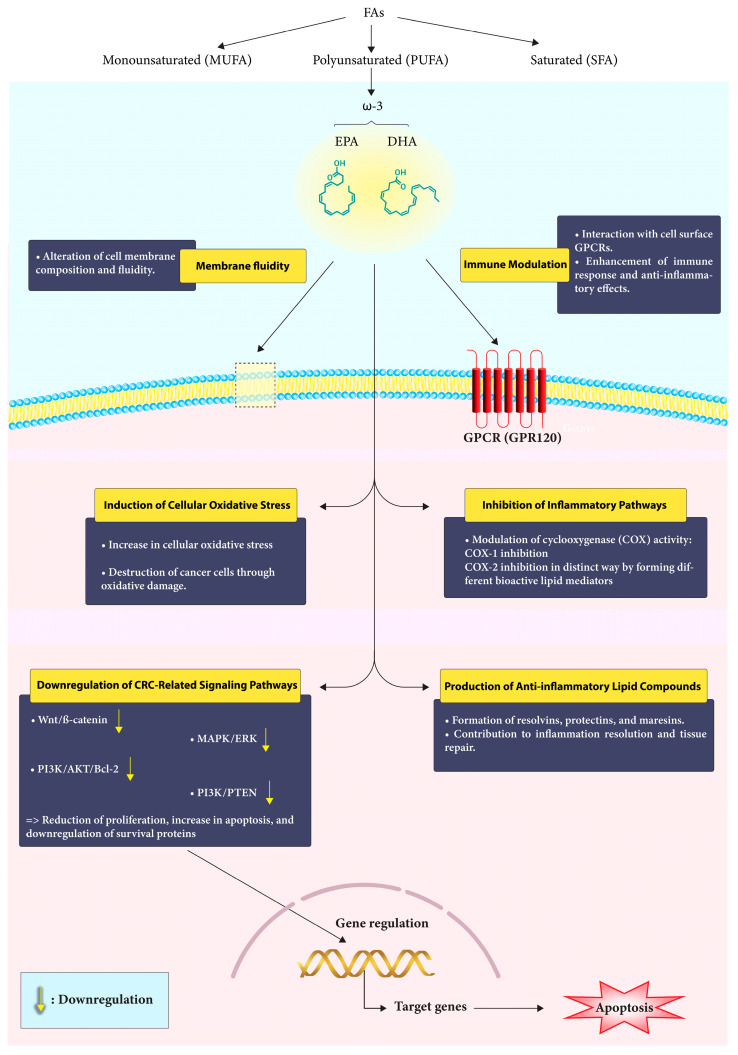
Mechanism of action of Omega-3 fatty acids in CRC. Omega-3 fatty acids engage with the cell membrane, modulate immune responses, and interfere with critical signaling pathways to exert their protective effects against colorectal cancer.

**Figure 2 cancers-15-04294-f002:**
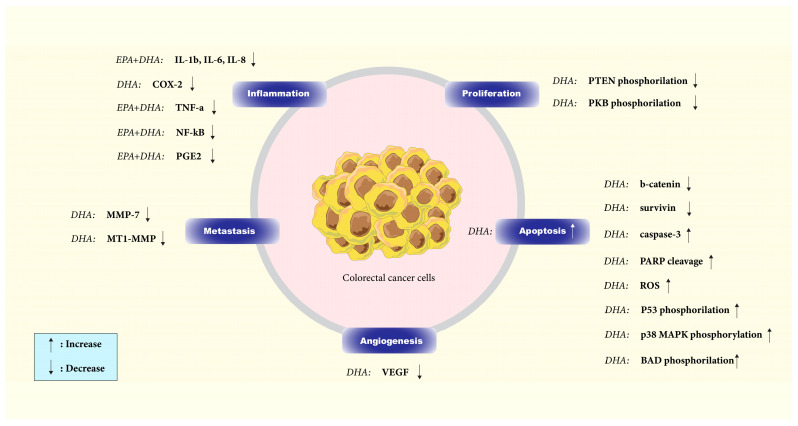
Effect of DHA and EPA on different components of signaling pathways in colorectal cancer. By inducing apoptosis and inhibiting proliferation, angiogenesis, and inflammation in colorectal cancer cells, DHA and EPA have demonstrated anticancer effects.

**Table 1 cancers-15-04294-t001:** Summary of preclinical and clinical trials on colorectal cancer using Ω3FAs.

Species	N	Ω3-FA Type	Combination	Chemo	Outcome	Study Reference
**Human**	10	EPA 4.1g/day + DHA 3.6 g/day (FO)	-	-	A remarkable decrease in the average percentage of replicative “S”-phase cells (a parameter in assessing the risk of colon cancer).	[67]
**Human**	476,160	EPA + DPA + DHA	-	NO	Ω3FA: ↓↓ risk of CRC	[69]
**Human**	68,109	EPA + DHA (FO)	-	NO	No association with CRC risk overall.	[51]
**Human**	1268	ω-3 PUFAs (EPA + DHA + DPA)	-	NO	ω-3 PUFAs: inversely related to the risk of cancer in the proximal site of the large bowel.	[70]
**Human**	18,682	EPA, DPA, DHA, LA, ALA	-	NO	− Shorter-chain PUFAs (LA, ALA): ↓CRC risk − Longer-chain PUFAs (AA, EPA, DPA): ↑ CRC risk.	[71]
**Human**	48,233	DHA	-	NO	− No overall association with CRC risk − High intake of fish-derived DHA: ↓ risk of rectal cancer.	[72]
**Human**	23	EPA + DHA (FO)	-	5-FU + irinotecan + folinic acid	− Clinically relevant decrease in the CRP/albumin relation. − Better efficacy + tolerability of chemotherapy.	[73]
**Human**	11	EPA + DHA (FO)	-	Capecitabine + Oxaliplatin + 5-FU + leucovorin	Improved CRP values, CRP/albumin status, plasma fatty acid profile, and potentially prevented weight loss during treatment.	[74]
**Human**	38	EPA	-	-	Elevated AA (Omega-6)/EPA ratio represents an inflammatory biomarker in tumor tissue of metastatic CRC	[75]
**Human**	709	EPA	+ Aspirin	-	No reduction CRC risk	[76]
**Human**	81	ω-3 capsule	+ Vitamin D	YES	IN CRC PATIENTS: co-supplementation ↓↓ inflammatory markers and CEA.	[77]
**Fat-1 TG mice**	-	DHA	-	-	DHA: ↑ apoptosis and ↓ tumor growth.	[78]
**Human**	123,529	Total omega-3 PUFA (ALA + EPA + DHA + DPA) Marine omega-3 PUFAs (EPA + DHA + DPA) Total omega-6 PUFAs (LA + AA)	-	-	Fish and marine ω-3 PUFA intake: ↓ risk of distal colon cancer and rectal cancer (men only).	[79]
**Human**	148	ω-3 FA–enriched ONS: 2.0 g EPA + 1.0 g DHA	-	-	Potential beneficial effect on local immune function	[80]
**Human**	148	ω-3 FA–enriched ONS: 2.0 g EPA + 1.0 g DHA	-	-	↓↓ production of LTB5 and 5-HEPE. ↓↓ production of LTB4 ↑↑anti-inflammatory effects in the surgical patient.	[81]
**Human**	148	ω-3 FA–enriched ONS: 2.0 g EPA + 1.0 g DHA	-	-	Infectious or non-infectious postoperative complications: no differences.	[82]
**Human**	30	EPA + DHA (FO)	-	YES	↑↑ tumor progression time.	[83]
**Human**	88	EPA-FFA (FFA)	-	-	Safe + well tolerated Preoperative FFA: ↑↑ postoperative OS and DFS.	[64]

Abbreviations: N, sample size; CRC, colorectal cancer; CRP, C-reactive protein; LTB5, leukotriene B5; LTB4, leukotriene B4; 5-HEPE, 5-hydroxyeicosapentaenoic acid; 5-HETE, 5-hydroxyeicosatetraenoic acid; EPA, eicosapentaenoic acid; DHA, docosahexaenoic acid; IL-6, interleukin-6; ONS, ω-3 FA-enriched oral nutrition supplement; FO, fish oil; OS, overall survival; DFS, disease-free survival; FFA, free fatty acid; ↑, indicate an increase; ↓, indicate an decrease; ↑ ↑, indicate a significant increase; ↓↓, indicate a significant decrease.

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
