# Peer review of "Anti-Inflammatory and Immune Properties of Polyunsaturated Fatty Acids (PUFAs) and Their Impact on Colorectal Cancer (CRC) Prevention and Treatment"

_cancers, 2023, doi:10.3390/cancers15174294_

Round 1

Reviewer 1 Report

Tojjari et al. gave an in-depth review about the effects of polyunsaturated fatty acids on preventing colorectal cancer and how it may impact treatment outcomes in colorectal cancer patients. With advances in CRC research, it is evident that modern lifestyles choices, including dietary, is becoming one of the major risk factors of colorectal cancer. It is therefore a timely review by the authors to address current knowledge and research gaps in the field on the topic of how dietary intake, with a focus on the different kinds of fatty acids, impacts the development and treatment of CRC.

From sections 1 through 3, the authors gave a good introduction on colorectal cancer and PUFAs and prefaced the importance of how different micro and macro nutrients, including fatty acids, influence CRC development and treatment outcomes. The authors delved deeper into potential roles of omega 3 FAs in impacting the risk of CRC occurrence through various mechanisms in sections 4 and 5. In section 6, the authors looked at clinical benefits and limitations of omega 3 FA supplementation to CRC patients during various treatment interventions such as chemotherapy and surgery. Section 7 describes literature on how omega 3 FAs might regulate the immune system to exert anti-cancer effects, and section 8 talks about the potential side effects of omega 3 supplementation.

Overall, the manuscript is a pleasant read, and I believe it is in line with the scope and of appeal to a broad audience of the journal. It is well balanced with discussions on the benefits of omega-3 fatty acids in various contexts, while also presenting potential limitations or lack of a beneficial response on the use of these FAs. The literature used within the review article are relatively new (most are with the past decade) which also includes clinical trials. Conclusions and discussion points drawn from the literature used are accurate, detailed and sometimes thought provoking. The authors pointed out an important aspect in their discussions that the field has yet to establish- given the numerous positive observed benefits, why has there not been concerted effort to research on and establish a consensus guideline on how best to complement omega 3 FAs at therapeutic doses with cancer treatment? It will also be interesting to see if there are studies that are able to identify a sub-group of CRC patients who do not benefit from the supplementation of omega 3 FAs during treatment, and possible ways to circumvent the issue. I think the manuscript is suitable for publication with the journal after some minor revisions:

Line 344: … reducing postoperative complications (Reference?) …

Please provide relevant reference(s) on studies that investigated the supplementation of omega 3 PUFA and improved pancreatic cancer patient outcomes.

Line 611: Missing reference

Section 7: I think it will be great if the authors could include some discussions on current literature about omega 3 FAs and immunotherapy.

Reviewer 2 Report

The review article “Anti-inflammatory and immune properties of Polyunsaturated Fatty Acids (PUFA) and their impact on Colorectal Cancer (CRC) prevention and treatment” is an interesting compilation of clinical studies in the area of dietary fatty acids and colorectal cancer. The authors have carefully compiled many relevant studies to summarize the current state of Ω3 FAs. The review is smoothly written and maintains interest while reading. With scholarly acumen, this manuscript describes the current landscape of scientific studies exploring ω-3 in colorectal cancer.

I have the following suggestions to improve it further:

1.     The role of probiotics appears out of context in paragraph 6.3 Ω3 FAs: an adjunct to chemotherapy.

2.     In paragraph 6.2, Cancer cachexia, missing reference text must be fixed.

3.     In Table 1, Summary of preclinical and clinical trials on colorectal cancer using Ω3FAs, I suggest reformatting the table and moving the ref column at the end. Also, in the current format, some of the text is broken between 2 lines, making it difficult to read. This, I assume anyway, will be corrected during later formatting.

4.     A combined figure to demonstrate all possible MOA of Omega FA will be helpful for the reader to receive a quick summary of the mechanisms of ω-3  FAs.

5.     The following appear to be important studies related to the review's focus and are missing in the current draft:

a.     “Effect of Supplementation With Marine ω-3 Fatty Acid on Risk of Colorectal Adenomas and Serrated Polyps in the US General Population. JAMA Oncol. 2020;6(1):108-115. doi:10.1001/jamaoncol.2019.4587.

b.     Effect of ω-3 fatty acids on rectal mucosal cell proliferation in subjects at risk for colon cancer. Gastroenterology, Volume 103, Issue 3, September 1992, Pages 883-891.
